# Influence of Laser Power on Microstructure and Properties of Al-Si+Y$_2$O$_3$ Coating

Yali Gao [1,*] , Pengyong Lu [1], Sicheng Bai [1], Baolong Qin [2] and Dongdong Zhang [1,*]

1 Department of Mechanical Engineering, Northeast Electric Power University, No. 169 Changchun Road, Chuanying District, Jilin 132012, China

2 School of Energy and Power Engineering, Northeast Electric Power University, No. 169 Changchun Road, Chuanying District, Jilin 132012, China

* Correspondence: dehuigyl@126.com (Y.G.); zhangdongdong@neepu.edu.cn (D.Z.)

**Abstract:** Al-Si/7.5 wt.%Y$_2$O$_3$ coatings were prepared on Mg alloy with laser cladding to enhance the wear and corrosion resistance of substrate. The influence of laser power on the microstructure and properties of the coating were discussed. The results uncovered that the coatings consisted primarily of Mg$_2$Si, Mg$_{17}$Al$_{12}$, Mg$_2$Al$_3$, Al$_4$MgY, and α-Mg phases. Through calculation, it was observed that the crystal size decreased with the decrease in the laser power. Y$_2$O$_3$ gave the coating a better strengthening effect due to the fine-grain strengthening and hard-phase strengthening. The average hardness of the coating with laser power of 1100 W achieved 312 HV, which was approximately 4.2 times that of the substrate. The wear volume of the coating was 22.2% that of the substrate. Compared with Mg alloy, the self-corrosion potential of the coating increased by 1.09 V, and the self-corrosion current density decreased by three orders of magnitude.

**Keywords:** Mg alloy; Al-Si alloy; laser power; microstructure; properties

## 1. Introduction

As one of the most abundant lightweight metal structural materials on the earth [1], Mg alloy is widely used in transportation, electronics, and energy fields due to its high specific strength and specific stiffness [2,3], good castability [4], excellent electromagnetic anti-interference ability, and high theoretical specific capacity of batteries [5,6] and is known as "the most promising green engineering structural material in the 21st century" [7]. However, the low hardness and poor corrosion resistance of magnesium alloy seriously limit its further application in industry.

Surface modification technology not only improves the surface properties of the materials but also maintains the original excellent properties of the substrate [8,9]. Laser-cladding technology has been widely applied in the surface processing of magnesium alloys due to its advantages such as high energy density, small heat-affected zone, and high bonding strength [10]. The physicochemical compatibility between aluminum and magnesium was observed to be good [11,12]. Therefore, laser cladding aluminum alloy onto magnesium obtains high bonding strength [13] and corrosion resistance [14,15].

Al-Si alloy has good physicochemical compatibility with magnesium alloy, which generates a variety of metal compounds to improve the performance of the coating. Therefore, researchers are committed to preparing higher-performance Al-Si coating on a magnesium matrix. Many scholars such as Rolink [16], Zhang [17], and Wang [18] have prepared Al-Si coatings on magnesium alloys. They explored the effects of laser scanning speed, laser power, and process angle on Al-Si cladding layers. It was found that the content of Mg$_{17}$Al$_{12}$, Mg$_2$Si, and Al$_3$Mg$_2$ was the highest when laser cladding was carried out with appropriate parameters, which leads to the coating having higher hardness and better corrosion resistance as compared with Mg alloy.

Rare earth (RE) elements have large atomic radius and strong chemical activity, which is easy to react with other elements. Appropriate addition of RE elements and their oxides refines the grains, purifies the structure, and reduces cracks, which improves the properties such as the hardness, corrosion resistance, and wear resistance of the coating [19,20]. Sun et al. [21] prepared Al-Cu coating containing $La_2O_3$ on AZ91D magnesium alloy via laser cladding. The result showed that the coating formed new strengthening phases ($LaAl_3$ and $Mg_{17}La_2$). Because the nucleation particles were increased and the microstructure was evenly distributed, the hardness and wear resistance of the coating was significantly higher than that of the substrate. Yang et al. [22] carried out laser cladding (Al + Ti + $B_4C$)/$Y_2O_3$ on AZ91D and found that the addition of rare earth elements produced more precipitated phases with different forms, which improved the properties of the coating. Zhang et al. [23] also found that proper $Nd_2O_3$ additions not only refined the crystalline structure and purified the impurities but also increased the amorphous content of the coating.

To sum up, it was found that there are few studies on the effect of rare earth elements on the properties of Al-Si coating. Previously, the effect of different $Y_2O_3$ content on the microstructure and properties of laser cladding Al-Si coating on AZ91HP was studied by our group [24]. It was found that the coating had the highest hardness and best wear resistance when $Y_2O_3$ content was 7.5 wt.%. In order to further study the change characteristics of laser energy density on the microstructure and properties of Al-Si/7.5 wt.% $Y_2O_3$, laser cladding was carried out with different laser power. The microstructure and properties of the coating were systematically analyzed.

## 2. Materials and Methods

### 2.1. Materials Used

In this study, AZ91HP die-casting magnesium alloy was used as the substrate, and its chemical composition is listed in Table 1. The samples of 50 mm × 30 mm × 10 mm were applied and polished with 600 mesh sandpaper to remove the oxide layer and reduce laser reflectivity.

**Table 1.** Chemical composition of AZ91HP (wt.%).

| Element | Al | Zn | Mn | Si | Fe | Cu | Ni | Be | Mg |
|---------|------|------|------|------|------|------|------|------|------|
| Composition (wt.%) | 8.8900 | 0.5620 | 0.2041 | 0.0443 | 0.0030 | 0.0034 | 0.0090 | 0.0012 | Bal. |

Al-Si eutectic powder (mass ratio 88:12) and $Y_2O_3$ were selected as cladding materials, in which $Y_2O_3$ content was 7.5 wt.%. Al-Si and $Y_2O_3$ powder were fully mixed and stirred in a ball mill. In the experiment, Al-Si eutectic powder (99.9% purity) was spherical, with a particle size of 48–147 μm. $Y_2O_3$ powder (9.9%purity) was granular, with an average particle size of 147 μm. SEM morphology of the two powders is shown in Figure 1.

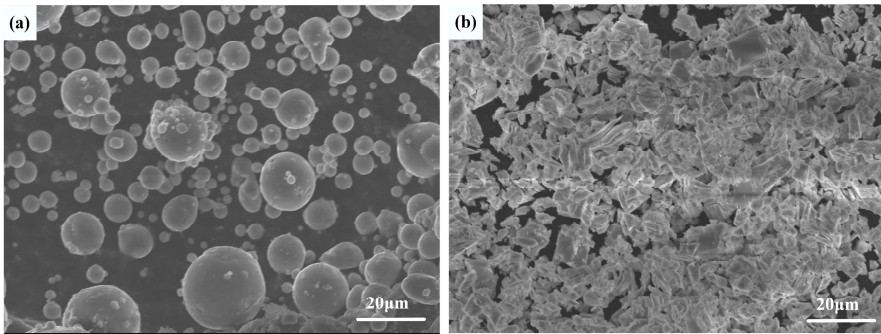

**Figure 1.** Scanning electron microscopic morphology of cladding powders: (**a**) Al-Si eutectic powder, (**b**) $Y_2O_3$.

### 2.2. Laser Machining

The laser-cladding equipment adopted was a DL-HL-T2000 helium-free cross-flow $CO_2$ laser (Shenyang Continental Laser Complete Equipment Co., Ltd., Shenyang, China). Its working principle is shown in Figure 2. The Al-Si/7.5 wt.%$Y_2O_3$ powder was preplaced on the substrate with a thickness of 1mm before laser cladding. Argon was used as a protective gas at a flow rate of 5 L/min. The experimental parameters are shown in Table 2.

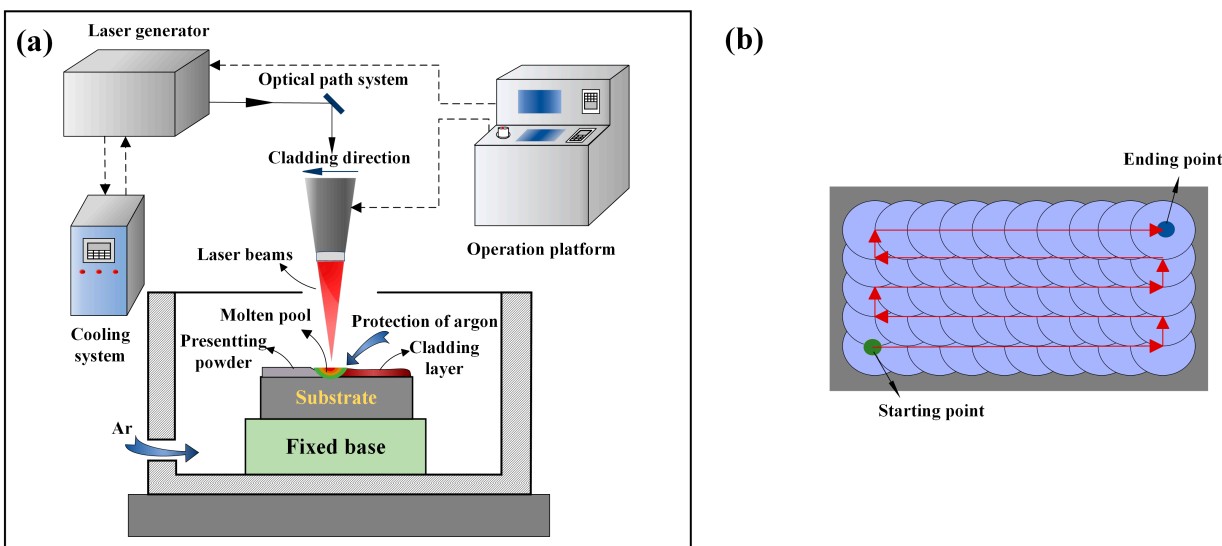

**Figure 2.** Principle diagram of the laser-cladding process: (**a**) laser processing process; (**b**) scanning path.

**Table 2.** Experimental parameters for laser-cladding coatings.

| Specimen Group | Laser Power (W) | Scanning Speed (mm/min) | Overlap Rate (%) | Spot Diameter (mm) |
|---|---|---|---|---|
| 1 | 1100 | | | |
| 2 | 1200 | 400 | 30 | 3 |
| 3 | 1300 | | | |
| 4 | 1400 | | | |

### 2.3. Microstructure and Properties Analysis

The coatings were polished and then subjected to X-ray diffraction (Dandong Tongda Technology Co., Ltd., Dandong, China) with Cu Kα radiation at 400 kV and 200 mA. The scanning speed was 1.020°/min, and the diffraction angle was 20°–80°. The microstructure of the coating was investigated using a JSM-7610 SEM, (Shanghai Baihe Instrument Technology Co., Ltd, Shanghai, China).

The microhardness of the coatings was measured with a Vickers tester (HXD-1000TMC/LCD, Innovatest, Maastricht, The Netherlands) with a load of 25 g for 10 s, while the space of two measured points was 0.05 mm.

High-speed reciprocating fatigue friction and a wear testing machine (MGW-02, Yihua, Jinan, China) was utilized to carry out the friction experiment. The friction ball pair was GCr15 steel with a diameter of 6.5 mm. The testing parameters were as follows: load 10 N, experimental frequency 10 HZ, sliding time 20 min, and reciprocal sliding distance 3 cm. The saturated calomel electrode was used as the reference electrode, and the three-electrode method was used to measure the potentiodynamic polarization. The electrochemical test was carried out in a 3.5 wt.% NaCl aqueous solution at room temperature.

## 3. Results and Discussion

### 3.1. Macroscopic Morphology

Figure 3 depicts the macroscopic photographs of the coatings with different laser power. Because $Y_2O_3$ improves the laser absorption rate of the powder and enhances the fluidity of the molten pool, the crystal morphology of the coating is obviously improved. However, the fish-scale-like trajectory of the coating is discontinuous because the lower energy density leads to the insufficient melting of the powder (Figure 3a). The surface of the coating is relatively smooth and well-formed when the laser power is 1200 W (Figure 3b). As is shown in Figure 3c,d, the collapse occurs in the later stage of laser cladding due to the excessive heat accumulation when the laser power increases to 1300 W and 1400 W.

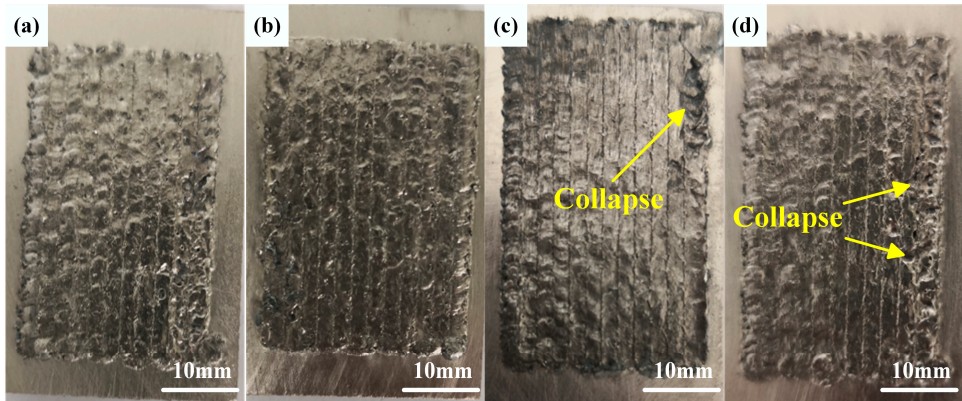

**Figure 3.** Surface morphology of coatings with different laser power: (**a**) 1100 W, (**b**) 1200 W, (**c**) 1300 W, (**d**) 1400 W.

### 3.2. Phase Composition

Figure 4 shows the XRD spectra of the coating with different laser power. The coatings are composed of $Mg_2Si$, $Mg_{17}Al_{12}$, $Mg_2Al_3$, $Al_4MgY$, and $\alpha$-Mg. The difference of thermal properties between Mg and Al is small. Therefore, part of the Mg alloy melted and penetrated into the coating to form the magnesium intermetallic compound. This ensures a strong metallurgical bonding between the substrate and the coating.

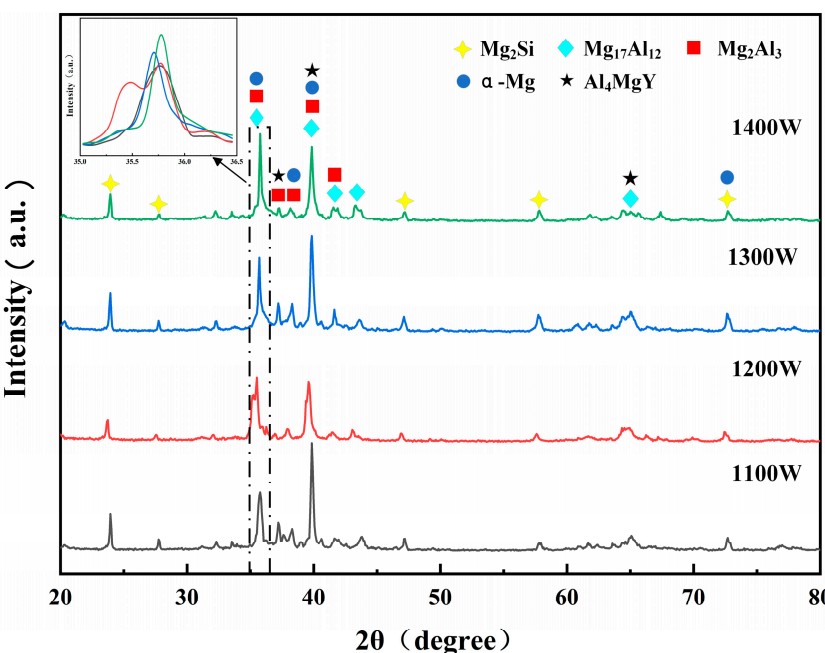

**Figure 4.** XRD spectra of the coatings with different laser power.

It can be seen from the inset of Figure 4 that the diffraction intensity of the peak at the angle of 35°–36° increases with the increase in laser power. The average grain size is roughly derived for the diffraction peak at the angle of 35°–36° by following Scherrer formula (Formula (1)) [25], which is listed in Table 3. The result demonstrates that the grain size gradually becomes larger with the increase in laser power. According to the analysis of the crystallization process, the increase in laser power leads to a reduction in the degree of undercooling of coating crystallization, which results in the increase in the grain size.

$$D = \frac{K\lambda}{\beta \cos \theta} \tag{1}$$

where $D$ represents the grain size, $\beta$ represents the half-peak width, $K$ is the Scherrer constant, and $\lambda$ denotes the X-ray wavelength with a given value of 1.54056 Å. $\theta$ is the Bragg diffraction angle.

**Table 3.** Grain sizes derived by Scherrer formula of different laser power.

| Laser power (W) | 1100 | 1200 | 1300 | 1400 |
|---|---|---|---|---|
| Half-peak width | 0.576 | 0.570 | 0.533 | 0.485 |
| Grain size (nm) | 15.2 | 15.3 | 16.4 | 18.0 |

### 3.3. Cross-Section Morphology

The cross-sectional morphologies of coatings are displayed in Figure 5. The coating, which has a thickness range of 0.7–1.0 mm, consists of three zones: the heat-affected zone (HAZ), bonding zone (BZ), and cladding zone (CZ). There exists a good metallurgical bonding between the coating and substrate. At a laser power of 1100 W, pores appear in the coating because the sublimated gas cannot volatilize into the environment before the solidification of the molten pool (Figure 5a). However, at laser power of 1200 W and 1300 W, the strong effect of Marangoni convection induces the gas to escape quickly, which makes the microstructure dense (Figure 5b,c). When laser power increases to 1400 W, the thermal stress increases significantly due to the increase in the temperature gradient, which leads to the formation of the crack (Figure 5d).

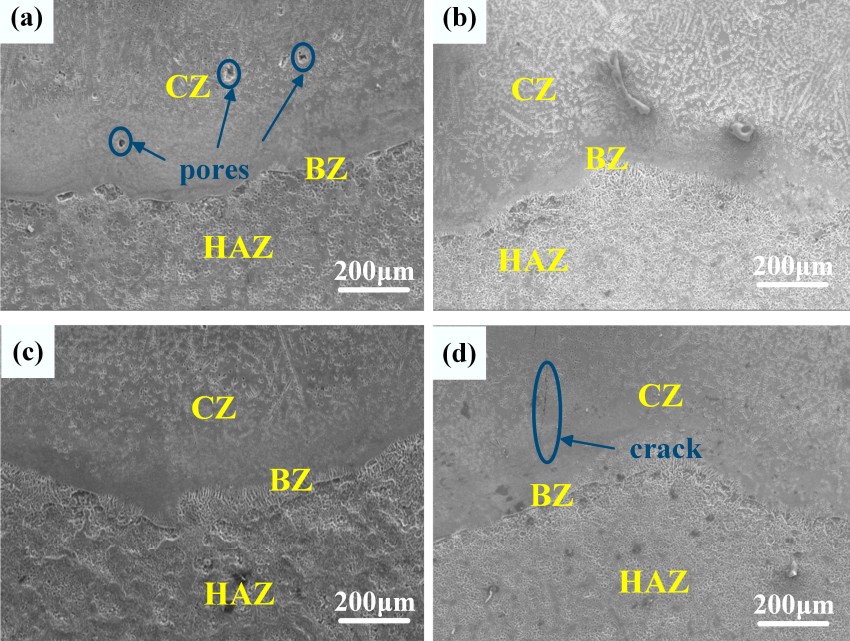

**Figure 5.** Cross-section morphologies of coatings with different laser power: (**a**) 1100 W, (**b**) 1200 W, (**c**) 1300 W, (**d**) 1400 W.

Figure 6 exhibits the distribution of Y, O, Si, Mg, and Al along the depth with different laser power. Figure 6a shows that the content of Mg in the coating is lower than that of the Al element at the lower laser power. The content of Mg and Al elements is flat when the power is 1200 W (Figure 6b). When the power increases to 1300 W, the laser energy density increases, and the substrate absorbs more energy. As is shown in Figure 6c, the dilution rate of the coating increases because of the enhanced fluidity of the molten pool, which results in the content of Mg element exceeding that of the Al element. When the power increases to 1400 W, the laser energy density is too high, resulting in some Mg elements being burned, so the Mg content in the coating decreases (Figure 6d).

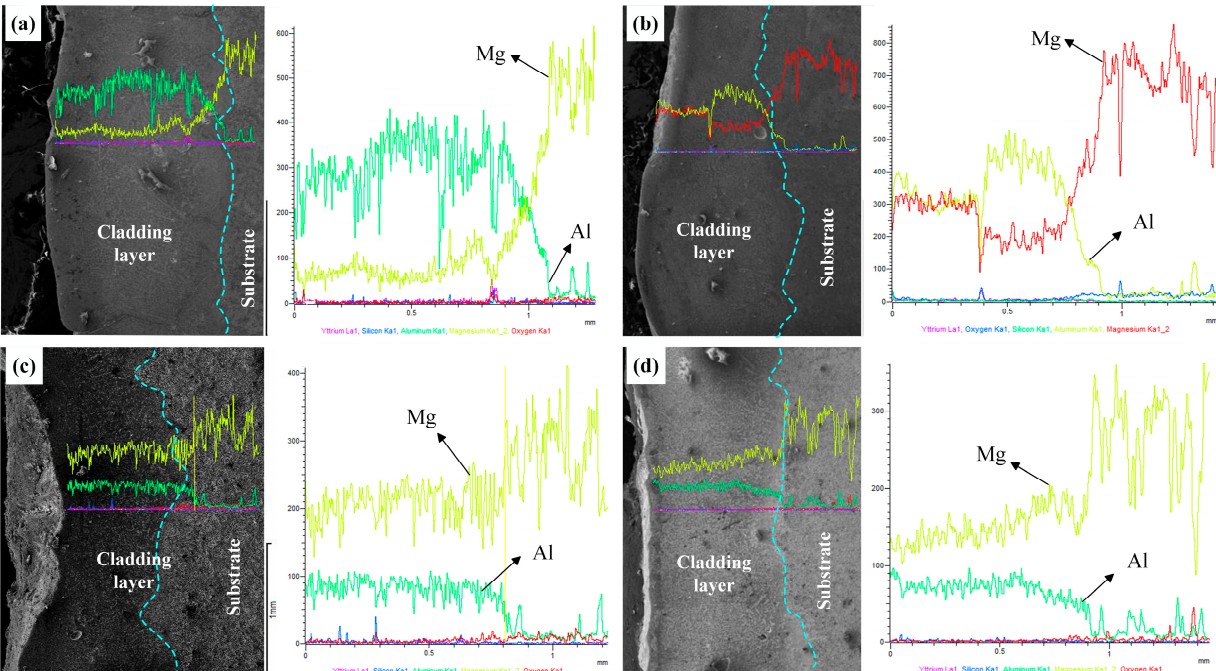

**Figure 6.** Cross-section line scanning results of different laser power coatings: (**a**) 1100 W; (**b**) 1200 W; (**c**) 1300 W; (**d**) 1400 W.

### 3.4. Microhardness Analysis

The microhardness distribution curve of the cross-section in the coating is exhibited in Figure 7. It can be seen that the hardness of the coatings is about 3.4–4.2 times that of the substrate (75 HV). The higher hardness of the coating is caused by a combination of fine-grain strengthening, hard-phase strengthening, and solid-solution strengthening. Firstly, laser cladding enhances the cooling rate of the melt, resulting in improved grain refinement. Additionally, the presence of $Y_2O_3$ helps to restrict grain growth and enhance nucleation rate, further contributing to the refinement of grains. Secondly, the elemental inter-diffusion between the coating and substrate results in the formation of a variety of high-hardness inter-metallic compounds. Thirdly, a rare earth atom enters the metal lattice through a vacancy diffusion mechanism and produces lattice distortion, which increases the resistance of the deformation and improves the strength and hardness of the material.

It is observed from Figure 7 that the hardness of the coating exhibits slight variations when the laser power is set within the range of 1100–1300 W (302–312 HV). However, the hardness of the coating decreases to 253.3 HV with the laser power increased to 1400 W. The reason is that the fine-grain strengthening effect is weakened due to the decrease in the cooling rate. At the same time, high energy density leads to the higher dilution rate of the coating, causing the formation of large amounts of α-Mg with the low hardness in the coating. Additionally, high energy density results in over-burning of the elements, reducing the density of the coating.

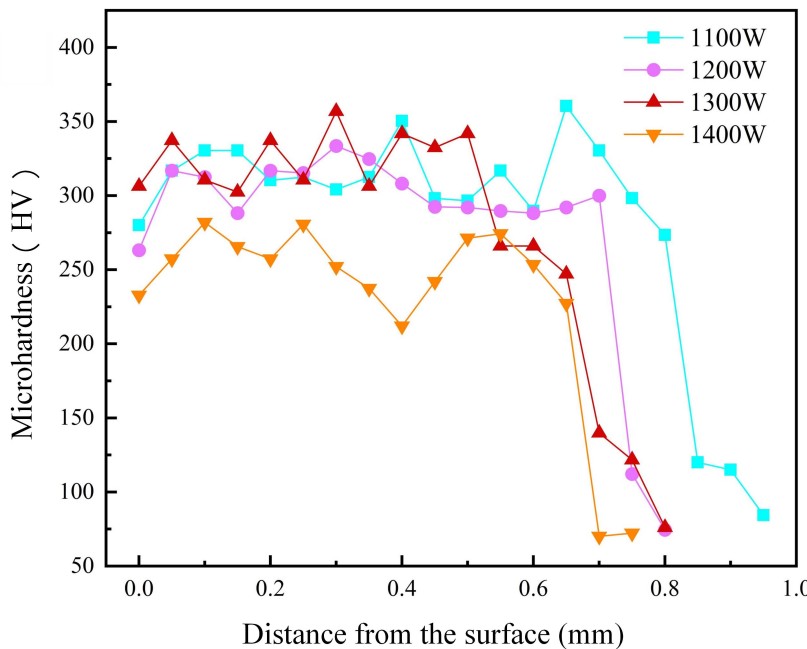

**Figure 7.** Microhardness distribution of coatings with different laser power.

*3.5. Wear Resistance*

The wear volume of the coating is shown in Figure 8. Generally, the higher hardness results in better wear resistance, so the wear volume of the coatings is 22.2%–41.5% that of the substrate. In addition, when the wear experiment begins, the grinding ball grinds the softer part of the coating, and the refined grains are staggered and dispersed, playing a good skeleton support role. The grains become coarser, and the supporting effect of the fine grains is weakened with the increase in laser power. Therefore, the wear volume of the coatings increases with the increase in laser power.

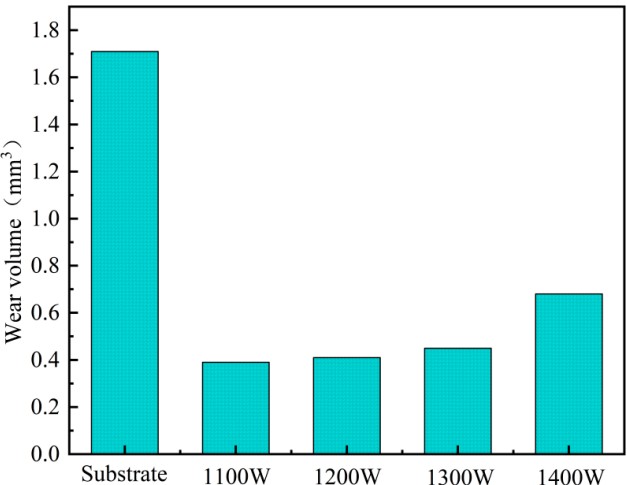

**Figure 8.** Wear volume of the substrate and coating with different laser power.

In order to further reveal the mechanism of the wear behavior, the SEM images of the wear surface are depicted in Figure 9. It can be observed from Figure 9a,b that there are many dense and deep parallel grooves and a little debris on the wear surface of the substrate, which indicates that the wear mechanism is abrasive wear. When the friction ball repeatedly slips on the substrate, severe plastic deformations occur when the normal stress and tangential stress are applied on the substrate. At the same time, a little debris with higher hardness falls off the substrate, which promotes the formation of the groove.

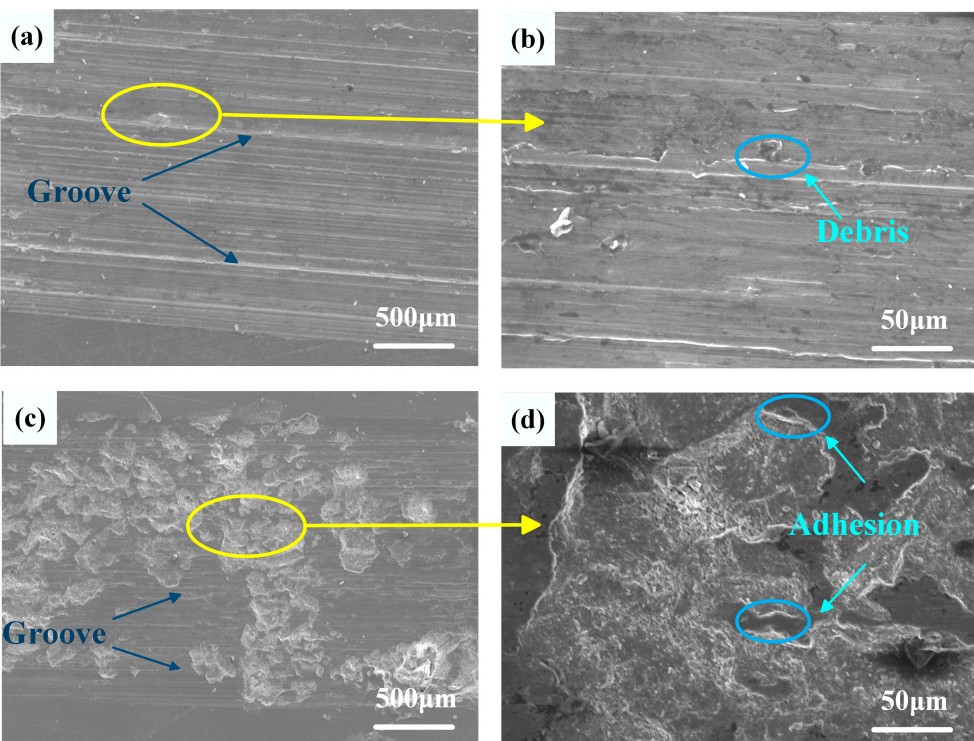

**Figure 9.** Friction track images of the substrate and coating by SEM: (**a**) and (**b**) the substrate, (**c**) and (**d**) the coating.

However, it is worth noting that Figure 9c seems to show that the groove is much shallower than that of the substrate because of the high microhardness of the coating. Additionally, typical adhesion can be seen in Figure 9c,d, which shows that wear mechanism of the coating exhibits adhesive wear besides the abrasive wear. In the high-speed sliding wear process, the surface pressure of the contact peak point reaches 5000 Mpa, which leads to the instantaneous temperature of the wear surface reaching 1000 °C. The adhesion is generated and destroyed in the subsequent sliding. Thus, adhesive wear is formed in the alternating processes of adhesion, wear, and re-adhesion [26].

*3.6. Corrosion Resistance*

It can be observed from Figure 10 that the open circuit voltage of the coatings is significantly higher than that of the substrate. In addition, the presence of typical passivation behavior is observed through the potentiodynamic polarization curve of the coating, as seen in the inset of Figure 10. A large amount of $Al_2O_3$ film is generated on the corrosion surface of the coating because of the addition of $Y_2O_3$, which covers the sample surface and precents the coating from being corroded further. Therefore, the current does not change significantly in spite of the increase in voltage. Furthermore, the slope of the cathodic polarization curve of is much higher than that of other coatings when the laser power is 1400 W, indicating that the corrosion resistance is very poor compared to that of a different laser power.

Table 4 summarizes the self-corrosion potentials and self-corrosion currents derived from the straight-line extrapolation of the Tafel cathode area in Figure 10. When the laser power is 1100 W and 1200 W, the self-corrosion current density is three orders of magnitude smaller than that of the substrate. The reasons are as follows: On the one hand, Figure 11a,b demonstrate that metallic compounds ($Mg_{17}Al_{12}$, $Mg_2Si$, $Mg_2Al_3$) with a lower potential difference in the coating replace the original reaction of α-Mg and $Mg_{17}Al_{12}$, forming new galvanic couples. On the other hand, rare earth atoms enrich at the grain boundary and act as a diffusion barrier for corrosion. Moreover, rare earth oxides prevent water molecules from penetrating into the corrosion boundary and inhibit the diffusion of gases outwards.

At the same time, the grain refinement reduces the effective area of the anode and cathode, which also greatly improves the corrosion resistance of the coating.

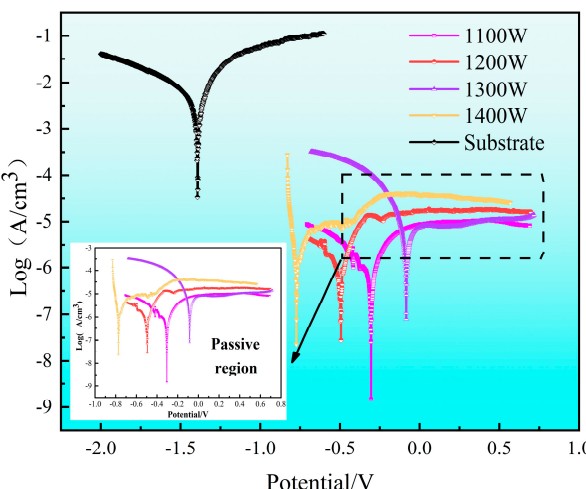

**Figure 10.** Polarization curves of the substrate and coatings with different laser power.

**Table 4.** Electrochemical parameters of coating and substrate in 3.5 wt.% NaCl solution.

| Sample | $E_{corr}$ (V) | $I_{corr}$ (A/cm$^2$) |
|---|---|---|
| Substrate | −1.394 | $4.168 \times 10^{-3}$ |
| 1100 W | −0.304 | $1.550 \times 10^{-6}$ |
| 1200 W | −0.494 | $2.173 \times 10^{-6}$ |
| 1300 W | −0.084 | $1.023 \times 10^{-5}$ |
| 1400 W | −0.774 | $1.661 \times 10^{-5}$ |

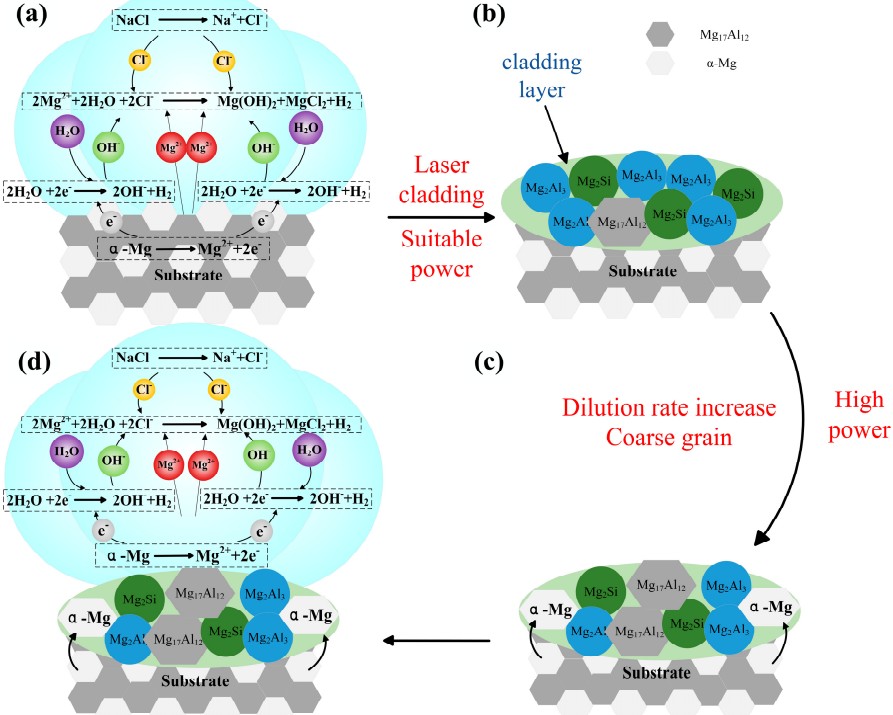

**Figure 11.** Schematic diagram of electrochemical corrosion: (**a**) electrochemical corrosion mechanism of substrate; (**b**) and (**c**) phase composition of coatings with different power; (**d**) electrochemical corrosion mechanism of coating.

However, the self-corrosion current density is only two orders of magnitude smaller than that of the substrate when the laser power increases to 1300 W and 1400 W. The reason why this corrosion resistance is worse than that of the lower laser power can be seen in Figure 11c,d. Firstly, a large amount of $\alpha$-Mg is generated because the dilution rate of the coating increases, replacing the intermetallic compound with a smaller potential difference as a new corrosion couple. Secondly, the grain becomes coarser, and the grain boundary distance increases, which results in the increase in the effective area for cathodic corrosion.

### 4. Conclusions

In this paper, the influence of laser power on Al-Si/7.5 wt.% $Y_2O_3$ coating on magnesium alloy was studied. The macroscopic morphology, microhardness, wear, and corrosion resistance of the coatings were analyzed. The following conclusions are drawn:

1.  All coatings are composed of $Mg_2Si$, $Mg_{17}Al_{12}$, $Mg_2Al_3$, $Al_4MgY$, and $\alpha$-Mg. The addition of $Y_2O_3$ increases the nucleation rate of the crystallization, contributing to refining the microstructure. Moreover, it is calculated that the grain size of the coating gradually becomes smaller as the laser power decreases.
2.  Compared to the substrate, the hardness of the coating is 3.4–4.2 times than it due to fine-grain strengthening, solid-solution strengthening, and hard-phase strengthening. Nevertheless, at the laser power of 1400 W, the hardness of the coating is lower due to the weakening of the fine-grain strengthening and hard-phase strengthening.
3.  The wear volume of the coating is 22.2%–41.5% that of the substrate. The increase in laser power leads to the coarsening of grains and the loss of fine-grain support during wear, resulting in the increase in wear volume. The wear surface of the coating present a composite wear form of abrasive wear and adhesive wear.
4.  The corrosion resistance of the $Y_2O_3$-modified sample is significantly better than that of the substrate because the surface of the coating generates a large amount of $Al_2O_3$ film and forms metallic compounds with a lower potential difference. The corrosion current density of the coating is the best when the laser power is 1100 W, which is three orders lower than the substrate. Moreover, the corrosion current density of the coating decreases with the increase in laser power.

**Author Contributions:** Conceptualization and writing—review and editing, Y.G.; writing—original draft preparation, P.L.; software, S.B.; resources, B.Q.; formal analysis, D.Z. All authors have read and agreed to the published version of the manuscript.

**Funding:** This research was funded by "The 13th Five-Year" Science and Technology research in Jilin Province and the Project of Science and Technology Development of Science and Technology Bureau in Jilin, Grant Numbers "20220098KJ" and "201831785".

**Institutional Review Board Statement:** Not applicable.

**Informed Consent Statement:** Not applicable.

**Data Availability Statement:** Not applicable.

**Acknowledgments:** Special thanks to the laser Laboratory of Northeast Electric Power University for assisting in this study.

**Conflicts of Interest:** The authors declare no conflict of interest. The funders had no role in the design of the study; in the collection, analyses, or interpretation of data; in the writing of the manuscript; or in the decision to publish the results.

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
