# Peer review of "Influence of Laser Power on Microstructure and Properties of Al-Si+Y2O3 Coating"

_coatings, doi:10.3390/coatings13071289_

Round 1
Reviewer 1 Report
This paper examines the influence of laser power on the microstructure and properties of the coating and generally presents a good methodological approach. The manuscript is original and could be of value to the scientific community therefore I support publication in COATINGS but after minor revision according to my queries.
(1) The abstract needs to be rewritten. It should be more concise. Probably should rewrite this part so that the methods are reflected in the results that then support the conclusions.
(2) I would like more details on the applied laser system in the section Materials and Methods. More details about laser would be useful for people who try to implement this technique.
(3) There are four tables in the manuscript, however, two tables are labeled with the name Table 3. Also, it is necessary to make changes in the text accordingly
The manuscript needs a thorough revision in terms of language. I understand the authors are not native English speakers but still, the manuscript should be written in a clearer way
Author Response
- The abstract needs to be rewritten. It should be more concise. Probably should rewrite this part so that the methods are reflected in the results that then support the conclusions.
Response 1. This part of the article has been amended
- I would like more details on the applied laser system in the section Materials and Methods. More details about laser would be useful for people who try to implement this technique.
Response 2. Thanks for your suggestion. This part of the article has been added in Figure 2 and on the line 69、70、71 and 82.
(3) There are four tables in the manuscript, however, two tables are labeled with the name Table 3. Also, it is necessary to make changes in the text accordingly
Response 3. This part of the article has been amended
Comments on the Quality of English Language
The manuscript needs a thorough revision in terms of language. I understand the authors are not native English speakers but still, the manuscript should be written in a clearer way
Response 4. Thanks for your suggestion. I made a lot of changes.
Reviewer 2 Report
The manuscript is not sound and doesn't provide anything new to the scientific community. In spite of that my comments on the manuscript are as follows:
1. What is the practical relevance or application scope of this work?
2. The absence of a clear justification for conducting this work at the end of the introduction.
3. In section 2.2, the explanation regarding the experimental procedure is lacking clarity and needs to be more explicit.
4. Is there a specific reason for considering a low load 5g (0.05N) for conducting hardness testing and wear tests in section 2.3.
5. Figure 2, 4, and 6 should include a clearly defined scale bar.
6. In section 3.2, it is worth noting that the calculation of grain size was based solely on measurements taken at ~35°, while there are peaks near ~40° that could potentially contribute to the overall grain size. This should be taken into consideration for a more comprehensive analysis.
7.In section 3.2, the explanation primarily focuses on grain size, overlooking the fact that the preferred orientation of coatings changes with power, as evident in Figure 3. It would be beneficial to include a discussion on the observed variations in preferred orientation and their potential implications.
8. After examining the XRD results, considering the inclusion of XPS or EDS analysis could enhance the quality of the paper and provide further insights into the composition of the coatings.
9. In section 3.4, on line 147, it is mentioned that the hardness of the coatings is about 3.1–3.7 times that of the substrate. However, there is no reference to the hardness values of the substrate in the paper or in figure 6.
10. What is the thickness of coatings.?
11. In addition to Figure 6, it would be beneficial to include a line graph or table depicting the hardness values of the samples for a clearer representation of the hardness values.
12. What is the reason behind the coatings exhibiting a lower wear volume compared to the substrate?
13.In section 3.6, a comprehensive explanation is provided considering Icorr values. However, it is important to note that in determining the corrosion resistance of coatings, Corrosion potential also plays a significant role in assessing the tendency of a metal to corrode. Unfortunately, this aspect is missing in this section.
14. line 199 A large amount of Al2O3 film is generated on the corrosion.... How do authors confirmed this?
The quality of the English language can be improved.
Author Response
- What is the practical relevance or application scope of this work?
Response 1. Thanks for your answer. This work aims to determine the wear and corrosion resistance of the Mg alloy surface, which will extend the service life of magnesium alloy parts. It was mentioned in lines 26,27.
- The absence of a clear justification for conducting this work at the end of the introduction.
Response 2. Thanks for your answer. At the end of the introduction, I introduced the experiment’s reason that explore the best process parameters of the coating powder. It was mentioned in lines 62-65.
- In section 2.2, the explanation regarding the experimental procedure is lacking clarity and needs to be more explicit.
Response 3. Thanks for your suggestion. This part of the article has been added in Figure 2 and on line 82.
- Is there a specific reason for considering a low load 5g (0.05N) for conducting hardness testing and wear tests in section 2.3.
Response 4. Thanks for your suggestion. It`s my fault to write the wrong date. This part of the article has been amended.
- Figure 2, 4, and 6 should include a clearly defined scale bar.
Response 5. Thanks for your suggestion. This part of the article has been amended.
- In section 3.2, it is worth noting that the calculation of grain size was based solely on measurements taken at ~35°, while there are peaks near ~40° that could potentially contribute to the overall grain size. This should be taken into consideration for a more comprehensive analysis.
Response 6. According to the literature, the Scherrer formula is generally applicable to the diffraction lines of low angle.
7. In section 3.2, the explanation primarily focuses on grain size, overlooking the fact that the preferred orientation of coatings changes with power, as evident in Figure 3. It would be beneficial to include a discussion on the observed variations in preferred orientation and their potential implications.
Response 7. In Section 3.2, paragraph 2, we find and state the effect of power on grain size on line 132-136.
- After examining the XRD results, considering the inclusion of XPS or EDS analysis could enhance the quality of the paper and provide further insights into the composition of the coatings.
Response 8. Thanks for your suggestion. This part of the article has been amended in the part 3.3.
- In section 3.4, on line 147, it is mentioned that the hardness of the coatings is about 3.1–3.7 times that of the substrate. However, there is no reference to the hardness values of the substrate in the paper or in figure 6
Response 9. Thanks for your suggestion. I have added the hardness value of the substrate on line 145.
- What is the thickness of coatings.
Response 10. The thickness of coatings is about 0.7-1.0mm, which was added on lines 143-145.
- In addition to Figure 6, it would be beneficial to include a line graph or table depicting the hardness values of the samples for a clearer representation of the hardness values.
Response 11. Thanks for your suggestion. This part of the article has been mentioned in the Figure 6.
- What is the reason behind the coatings exhibiting a lower wear volume compared to the substrate?
Response 12. The lower hardness is mainly affected by grain refinement and the higher hardness, and it was mentioned on line 177-183, which was marked with blue words.
13.In section 3.6, a comprehensive explanation is provided considering Icorr values. However, it is important to note that in determining the corrosion resistance of coatings, Corrosion potential also plays a significant role in assessing the tendency of a metal to corrode. Unfortunately, this aspect is missing in this section.
Response 13. Thanks for your suggestion. I learned that measuring the corrosion resistance is mainly based on the size of the corrosion current from the literature I've read.
- line 199 A large amount of Al2O3 film is generated on the corrosion.... How do authors confirmed this?
Response 14. In this part, the conclusion is based on our previous research in the literature.
Reviewer 3 Report
See comments below regarding English language usage. The authors must respond to the following comments and issues.
1. In figure 1 and subsequent microscopy results the text labeling and dimension labels for the micron markers b impossible to read and should be presented in a larger font.
2. In the materials and methods section the authors should provide more details of the experiments. For example, how was the substrate sample oriented - presumably the surface to be coated was horizontal in orientation and flat. Sufficient information should be provided so that this work could be replicated in another laboratory.
3. How thick was the coating? The intended and experimentally attained thicknesses should be stated and the labeling of the cross sections in Figure 4 should be improved. The hardness data in Figure 6 suggests a thickness of 0.6 to 0.8 mm was attained - was this intended?
4. The data, especially the grain sizes, in Table 3 are stated to three digits beyond the decimal point. Such accuracy is not possible.
The authors must seek the services of a competent English language editor. Usage is frequently incorrect and sometimes confusing. Non-English speakers may find this manuscript difficult to read in its present form.
Author Response
- In figure 1 and subsequent microscopy results the text labeling and dimension labels for the micron markers b impossible to read and should be presented in a larger font.
Response 1. Thanks for your suggestion. This part of the article has been amended.
- In the materials and methods section the authors should provide more details of the experiments. For example, how was the substrate sample oriented - presumably the surface to be coated was horizontal in orientation and flat. Sufficient information should be provided so that this work could be replicated in another laboratory.
Response 2. Thanks for your suggestion. This part of the article has been added in Figure 2 and on the line 82.
- How thick was the coating? The intended and experimentally attained thicknesses should be stated and the labeling of the cross sections in Figure 4 should be improved. The hardness data in Figure 6 suggests a thickness of 0.6 to 0.8 mm was attained - was this intended?
Response 3. The thickness of coatings is about 0.7-1.0mm, which was added on line 143-145 and was intended. And the Figure 4. has been amended.
- The data, especially the grain sizes, in Table 3 are stated to three digits beyond the decimal point. Such accuracy is not possible.
Response 4. Thanks for your suggestion. This part of the article has been amended.
Reviewer 4 Report
This article presents a highly commendable and captivating study that explores the impact of laser power on the microstructure and properties of Al-based coatings on Mg alloy. The abstract is exceptionally well-written, effectively highlighting the research's objectives and principal findings. The procedure is clearly outlined, and the abstract concisely summarizes the results within the context of the identified research problem. The study extensively reviews a wide range of literature, providing a solid justification for the importance of this research. Notably, it was observed that the coating's hardness experiences a significant improvement attributed to fine grain strengthening, solid solution strengthening, and hard phase strengthening. This conclusion is drawn based on rigorous experimental investigations. The reviewer recommends accepting this article pending the authors' addressing of the following concern.
In order to improve the clarity and transparency of the citations, it is recommended to avoid grouped citations such as [2-6], [11-15] etc. Instead, it would be more informative to individually list each reference within the citation. By providing specific citation numbers for each source, readers will have a clear understanding of which references have been cited and can easily locate the corresponding sources for further reading.
Table 2: Please combine the rows for scanning speed, overlap rate, and spot diameters, as these parameters are identical for all specimen groups.
Attention should be given to the presence of two instances of Table 3 in the manuscript. It is essential to ensure careful consideration of this issue. One possible solution is to merge the contents of both Table 3 instances with Table 2. By doing so, valuable space can be saved, and it will enhance readability for the reader, making it more convenient to navigate through the data.
The authors' work is brilliantly depicted in Figure 10, showcasing excellence in its representation. I commend the authors for their remarkable contribution and encourage them to continue their exceptional efforts.
Please include a scale bar in Figure 2.
Please improve figures 4 and 8, especially the scale bar and the magnification. Please increase the font size so that the level of magnification can be clearly identified.
Based on Figures 6 and 7, it is evident that a laser power of 1100 W yields the best results in terms of both wear characteristics and hardness profile. Notably, as the laser power increases beyond this point, both characteristics exhibit a decline. However, it remains unclear how the performance would be affected if the laser power were further reduced, for example, to 600 W. From an operational perspective, one might assume that performance would improve when decreasing the laser power from 1400 W. However, it is important to note that there is likely an optimal range, beyond which performance would start to deteriorate again.
This research does not provide an answer regarding the optimum laser power. The authors should either discuss the potential outcomes if the laser power is decreased below 1100 W or provide a rationale for selecting the specific range of laser power between 1100 W and 1400 W. Additional insights into these aspects would enhance the completeness of the study.
Author Response
In order to improve the clarity and transparency of the citations, it is recommended to avoid grouped citations such as [2-6], [11-15] etc. Instead, it would be more informative to individually list each reference within the citation. By providing specific citation numbers for each source, readers will have a clear understanding of which references have been cited and can easily locate the corresponding sources for further reading.
Response 1. Thanks for your suggestion. This part of the article has been amended.
Table 2: Please combine the rows for scanning speed, overlap rate, and spot diameters, as these parameters are identical for all specimen groups.
Response 2. Thanks for your suggestion. This part of the article has been amended.
Attention should be given to the presence of two instances of Table 3 in the manuscript. It is essential to ensure careful consideration of this issue. One possible solution is to merge the contents of both Table 3 instances with Table 2. By doing so, valuable space can be saved, and it will enhance readability for the reader, making it more convenient to navigate through the data.
Response 3. Thanks for your suggestion. This part of the article has been amended.
The authors' work is brilliantly depicted in Figure 10, showcasing excellence in its representation. I commend the authors for their remarkable contribution and encourage them to continue their exceptional efforts.
Response 4. Thanks for your affirmation.
Please include a scale bar in Figure 2.
Response 5. Thanks for your suggestion. This part of the article has been amended.
Please improve figures 4 and 8, especially the scale bar and the magnification. Please increase the font size so that the level of magnification can be clearly identified.
Response 6. Thanks for your suggestion. This part of the article has been amended.
Based on Figures 6 and 7, it is evident that a laser power of 1100 W yields the best results in terms of both wear characteristics and hardness profile. Notably, as the laser power increases beyond this point, both characteristics exhibit a decline. However, it remains unclear how the performance would be affected if the laser power were further reduced, for example, to 600 W. From an operational perspective, one might assume that performance would improve when decreasing the laser power from 1400 W. However, it is important to note that there is likely an optimal range, beyond which performance would start to deteriorate again.
This research does not provide an answer regarding the optimum laser power. The authors should either discuss the potential outcomes if the laser power is decreased below 1100 W or provide a rationale for selecting the specific range of laser power between 1100 W and 1400 W. Additional insights into these aspects would enhance the completeness of the study.
Response 7. Thanks for your suggestion. We have noticed this problem before writing this paper. But we did not add the analysis of low-power samples because the samples with the laser power of 900W and 800W are not well formed.
Round 2
Reviewer 2 Report
The manuscript has undergone significant improvements, courtesy of the authors' efforts. I have a suggestion regarding the conclusion. It would be more effective if the conclusion were presented in paragraph form rather than as bullet points. Additionally, it would be advantageous to include specific numerical data rather than solely mentioning that "The hardness of the coating is improved significantly due to fine grain strengthening."
Author Response
The manuscript has undergone significant improvements, courtesy of the authors' efforts. I have a suggestion regarding the conclusion. It would be more effective if the conclusion were presented in paragraph form rather than as bullet points.
Response 1. Thank you very much for your suggestion, and we have carefully considered it. Compared with the paragraph form, the bullet point form may be more clearly and simply state the conclusion of our experiment
Additionally, it would be advantageous to include specific numerical data rather than solely mentioning that "The hardness of the coating is improved significantly due to fine grain strengthening."
Response 2. Thank you very much for your suggestion. This part of the article has been amended
Reviewer 4 Report
Can be accepted now
Author Response
Thank you for your review.